# Development and Assessment of a Multiple-Analysis System for Diagnosing Malaria and Other Blood Parasite Infections in Humans and Non-Human Primates

**DOI:** 10.3390/diagnostics15050620

**Published:** 2025-03-04

**Authors:** Ángela Ceballos-Caro, Víctor Antón-Berenguer, Marta Lanza, Justinn Renelies-Hamilton, Amanda Barciela, Pamela C. Köster, David Carmena, María Flores-Chávez, Emeline Chanove, José Miguel Rubio

**Affiliations:** 1Parasitology Reference and Research Laboratory, National Microbiology Centre, Instituto de Salud Carlos III, 28220 Majadahonda, Spain; angieceba@hotmail.com (Á.C.-C.); victor.antonb@gmail.com (V.A.-B.); mlanza@isciii.es (M.L.); pamelakster@yahoo.com (P.C.K.); dacarmena@isciii.es (D.C.); mflores@isciii.es (M.F.-C.); 2Department of Microbiology and Parasitology, Hospital Universitario Severo Ochoa, 28914 Leganés, Spain; 3Section for Ecology and Evolution, Department of Biology, University of Copenhagen, 2200 Copenhagen, Denmark; claxon71@gmail.com; 4Jane Goodall Institute Spain in Senegal, Dindefelo Biological Station, Dindefelo 26005, Senegal; amanda.barciela@gmail.com; 5Women for Africa Foundation, 28046 Madrid, Spain; 6Faculty of Health Sciences, Alfonso X El Sabio University (UAX), 28091 Villanueva de la Cañada, Spain; 7CIBERINFEC, ISCIII—CIBER Infectious Diseases, Instituto de Salud Carlos III, 28029 Madrid, Spain; 8Clinique Vétérinaire du Val d’Arve, 74930 Reignier-Esery, France; emeline.chanove@veterinaire-valdarve.com

**Keywords:** molecular diagnosis, RT-PCR, malaria, Filariae, Trypanosomatidae, neglected tropical diseases

## Abstract

**Background/Objectives**: Many tropical diseases such as malaria, Chagas, human African Trypanosomiasis, and Lymphatic filariasis coexist in endemic countries, affecting more than 1 billion people worldwide, and are recognised as major global vector-borne diseases. Tackling this disease requires an accurate diagnosis that is sensitive, specific, and rapid. This study aimed to describe and validate a new highly sensitive and specific multiple-analysis system that can effectively detect numerous etiological agents in a single test. **Methods**: A total of 230 human blood samples were assessed retrospectively for parasite characterisation, as well as 58 stool samples from non-human primates. Primers and probes were designed in the small subunit ribosomal RNA gene, except for *Plasmodium* spp., for which the novel target was Cytochrome Oxidase Subunit 1. **Results**: The analytical specificity of the presented method was 100%, with no unspecific amplifications or cross-reactions with other blood parasitic diseases. The detection limit obtained was between 0.6 and 3.01 parasites/µL for *Plasmodium* species, 1.8 parasites/mL for Trypanosomatidae, and 2 microfilariae/mL in the case of Filariae. The sensitivity, specificity, predictive values, and kappa coefficient reached almost 100%, except for Filariae, whose sensitivity dropped to 93.9% and whose negative predicted value dropped to 89.5%. The operational features described a turnaround and a hands-on time shorter than the compared methods with a lower cost per essay. **Conclusions**: This work presents a cost-effective and highly sensitive multiplexed tool (RT-PCR-bp) capable of performing simultaneous detection for blood parasitic diseases using specific fluorescence probes, enabling the diagnosis of low parasite loads and coinfections.

## 1. Introduction

Many tropical diseases are caused by parasites and transmitted by vectors. Malaria is the most important, and it killed 608,000 people in 2022 [1]. Additionally, many of the so-called neglected tropical diseases (NTDs, i.e., lymphatic filariasis, onchocerciasis, Chagas disease, and human African trypanosomiasis) affect near 1.65 billion people globally [2,3]. The first step for appropriate control lies in a correct diagnosis, which, following World Health Organization (WHO) recommendations, should be sensitive, highly specific, and applicable in endemic areas with limited resources [2,4].

Tropical and subtropical areas of the planet harbour the bulk of parasitic infections. In these regions, coinfections are common due to high prevalence and infection/re-infection rates [3,5]. Control programmes for such diseases—including those targeting malaria, African trypanosomiasis, or other specific pathologies—are conducted independently [6,7], without tackling these diseases jointly to optimise available human and technical resources. In clinical cases, regardless of the setting (endemic or non-endemic), if the patient has fever, then malaria tests are performed, and other concomitant pathologies are only looked for if the result for malaria is negative, with some exceptions according to the experience of the clinician [8]. Molecular techniques have greatly contributed to improvements in the diagnosis of multiple diseases, including parasitic diseases. However, most available tests are directed towards individual pathogens, impairing the detection of other sympatric pathogens that can coinfect the same patient. To bridge this limitation, multiple-analysis systems (MASs) allow the detection of multiple pathogens in a single test and within a single sample, reducing screening costs and diagnostic turnaround times [9,10]. In addition, a wide range of MASs have been developed to test a variety of biological samples, including faecal matter, skin, saliva, urine, or blood, as well as for the analysis of genetic material in transmission vectors [11]. However, only a few MASs currently target NTDs.

Here, we describe an MAS based on real-time PCR (RT-PCR), specifically designed for the detection of specific groups of parasites (*Plasmodium* spp., Trypanosomatidae, and Filariae) that, at some stage of their life cycle, are present in blood. The method is carried out in a single tube using different primers and specific probes and incorporates an internal reaction control to distinguish between true negatives and potential amplification failures or suboptimal DNA extraction (false negatives). Primers and probes were designed in the small subunit ribosomal RNA (ssrRNA) gene, except for *Plasmodium* spp., for which the novel target was Cytochrome Oxidase Subunit 1 (COI), due to its higher sensitivity [12,13].

## 2. Materials and Methods

The developed method was validated by comparing the obtained results against accredited laboratory methods for detecting these parasites, including (i) nested multiplex PCR for *Plasmodium* spp. detection [14,15], (ii) nested PCR for Filariae detection [16], and (iii) RT-PCR for Trypanosomatidae detection [17]. The evaluation of the MAS was carried out following the recommendations of the Spanish Society of Infectious Diseases and Clinical Microbiology [18]. Performance parameters, including analytical sensitivity and specificity, positive and negative predictive values, analytical sensitivity, reproducibility, and repeatability, were estimated. In addition, we calculated the operational characteristics (turnaround time and economic cost).

### 2.1. Samples

Two sets of samples were retrospectively investigated in the present study:
(a)Anonymised blood samples (*n* = 230) from Spanish returning travellers or immigrants coming from Sub-Saharan Africa, India, Pakistan, and Venezuela were sent to the Parasitology Reference and Research Laboratory at the National Microbiology Centre-Instituto de Salud Carlos III for testing for malaria and other tropical diseases. These samples were part of the repository of the Spanish National Biobanks (Registry number: C.0001392), including the samples used in this study, which came from three research projects for the study of imported malaria in Spain, which were approved by the Ethics Committee of the Instituto de Salud Carlos III and the Research Ethics Committee of the 12 de Octubre University Hospital in Madrid (ISCIII CEI PI 74_2020 Date: 30 September 2020; ISCIII CEI PI 100_2022 Date: 26 January 2022; and H12O CEtm:.18/021 Date: 8 February 2022). They consisted of 119 *Plasmodium*-positive samples, including *P. falciparum* (*n* = 81), *P. viva*x (*n* = 13), *P. ovale* (*n* = 8), *P. malariae* (*n* = 8), and mixed infections by two *Plasmodium* species (*n* = 9). In addition, 33 samples for Filariae (*Loa loa*, *n* = 24; *Mansonella perstans*, *n* = 7; and mixed infections by *L. loa* + *M perstans*, *n* = 2), 9 samples for Trypanosomatidae (*Trypanosoma brucei*, *n* = 4; *Leishmania infantum*, *n* = 3; and *Trypanosoma cruzi*, *n* = 2), and 69 negative samples were also available for the survey.(b)We analysed faecal samples (*n* = 58) from wild chimpanzees (*Pan troglodytes verus*) collected in the Dindéfélo Community Nature Reserve (Senegal). These samples were a subset of the initial panel (*n* = 234) originally used to screen for the presence of intestinal and hematic parasites [17]. The animal study protocol was approved by the Research Ethics committee of the Instituto de Salud Carlos III (protocol code CEI PI 90_2018-v2), and the study was conducted in strict accordance with the Code of Best Practices for Field Primatology of the International Primatological Society [19,20]. They included 2 *Plasmodium*-positive samples (*P. malariae*, *n* = 1; *Plasmodium* spp., *n* = 1), 16 Trypanosomatidae-positive samples (*T. brucei* sp., *n* = 1; *Phytomonas* sp., *n* = 8; *Trypanosomatidae* spp., *n* = 5; *Bodo* sp., *n* = 1; *Neobodo* sp., *n* = 1), 9 Filariae-positive samples (*M. perstans*, *n* = 8; *Mansonella* spp., *n* = 1), and 31 negative samples. No experimentation was conducted on chimpanzees; all faecal samples were collected non-invasively from the ground without disrupting any wild animal.


### 2.2. DNA Extraction and Purification

Genomic DNA was extracted from 200 µL of whole-blood samples collected in EDTA and stored at −20 °C using the QIAamp DNA Mini Blood Kit (QIAGEN^®^, Hilden, Germany), according to the manufacturer’s instructions.

Genomic DNA samples stored at 4 °C were originally isolated from 200 mg of faecal samples preserved in 70% ethanol using the QIAamp DNA Stool Mini Kit (QIAGEN^®^, Hilden, Germany) according to the manufacturer’s instructions.

### 2.3. Primer and Probe Design

Three PCR primers, one forward and two reverse, were designed in the small subunit ribosomal RNA (ssrRNA) gene sequence to characterise the Trypanosomatidae and Filariae families. The forward primer hybridised with both families, whereas reverse primers were specific for each one. To identify *Plasmodium* spp., two new primers were designed based on the cytochrome oxidase subunit 1 (COI) gene sequence. For the internal reaction control, previously designed primers targeting the human ssrRNA were used [14,15]. All primers (Table 1) were designed in silico under different cycling conditions such as maximum specificity, alignment temperatures in the range of 60–62 °C, and a maximum GC base content between 40% and 60% without self-complementarity, among others [14,21,22]. Probes were designed at 69–72 °C with an annealing temperature 7 to 10 °C higher than that calculated for our primers [21,22]. The diagnostic performance of the designed primers was assessed in the laboratory with real human DNA samples.

### 2.4. Real-Time PCR for Blood Parasites (RT-PCR-bp) Groups

The RT-PCR-bp reaction mix consisted of 1x Quantimix HotSplit (Biotools, Madrid, Spain), which contained the buffer, the polymerase, and dNTPs, the corresponding amounts of primers and probes all mixed together in a single tube (Table 1), and 5 µL of template DNA in a final reaction volume of 20 µL.

The amplification conditions consisted of an initial denaturation step of five min at 95 °C, followed by 45 cycles of 10 s at 95 °C and 30 s at 60 °C, where fluorescence was read in the yellow (excitation peak, 533 nm; emission peak, 559 nm), orange (excitation peak, 596 nm; emission peak, 615 nm), red (an excitation peak at 651 nm and an emission peak at 670 nm), and crimson (excitation peak, 683 nm; emission peak, 703 nm) channels. Amplification was performed in a Rotor-Gene Q 6 plex (QIAGEN^®^, Hilden, Germany).

All samples were analysed in duplicate, and positive controls for each group of parasites, a known negative sample, and DNA and No-DNA isolation controls were added to each reaction to detect possible reagent contamination.

### 2.5. Validation

The validation of the method was performed by the direct comparison of the obtained results with those from accredited PCR protocols in the laboratory. The reference method for *Plasmodium* detection was a nested multiplex malaria PCR (SnM-PCR) that identified the four *Plasmodium* species that normally infect humans (*P. falciparum*, *P. malariae*, *P. ovale*, and *P. vivax*) [14,15]. The reference method for Filariae was a nested PCR (nFil-PCR) that identified, by the size of the amplified fragments, the majority of the Filariae including *Wuchereria bancrofti*, *Loa loa*, *Mansonella perstans*, *M. ozzardi*, *Onchocerca volvulus*, and *Dirofilaria* spp. [16]. The reference method for Trypanosomatidae was RT-PCR (RT-PCR-Tryp), capable of identifying *T. brucei*, *T. cruzi*, *Leishmania* spp. and other kinetoplasts [17]. In the case of discrepant results, both tests were repeated. The cases in which the IC was negative were not considered.

### 2.6. Analytical Sensitivity and Specificity

The analytical sensitivity (or limit of detection, LoD) was calculated in positive samples for each human *Plasmodium* species, one *T. brucei* sample, and one *L. loa* sample using 10-fold serial dilutions. All samples were tested in duplicate. The initial parasitaemia of samples was calculated by microscopy using Giemsa-stained blood smears and following WHO recommendations [23,24,25] (Table 2).

The LoD was defined as the lowest parasite concentration in which samples and their duplicates were positive [26]. The analytical specificity was determined from the blood samples that were positive for one of the blood parasite groups but negative for the others, detecting possible cross-reactions.

### 2.7. Intra-Assay Precision

To calculate the intra-assay precision (or repeatability) of the technique, three positive DNA samples for each group of blood parasites with three different levels of parasitaemia (high, medium, and low) were analysed by RT-PCR-bp on three consecutive days and using the same RT-PCR machine [18].

### 2.8. Inter-Assay Precision

To calculate the inter-assay precision (or reproducibility) of the technique, amplifications of three positive DNA samples for each group of blood parasites with three different levels of parasitaemia (high, medium, and low) were carried out on the same day using three different RT-PCR machines [18].

### 2.9. Statistical Analysis

Performance parameters including sensitivity, specificity, positive predictive value (PPV), negative predictive value (NPV), accuracy, and kappa index were calculated with 95% confidence intervals (95% CIs) using Epi Data software version 3.1.

### 2.10. Operational Features

The processing time was measured from the moment that the sample was processed for nucleic acid extraction until the diagnostic result was obtained. Costs per sample were calculated, excluding the expenses for controls included in each run and any duplication of samples. Staff-related costs were estimated based on the time required to perform the techniques. These cost estimates were specific to procedures conducted in Spain; in other countries, kit prices may vary significantly, even between institutions. However, the relative cost differences are expected to remain comparable.

## 3. Results

### 3.1. Validation

The primer set concentrations used in RT-PCR-bp were calculated in silico according to the temperature. The optimal binding temperature for each set of primers was determined using gradient PCRs in a C1000 Thermal Cycler (BioRad, Berkeley, CA, USA). This temperature was set at 60 °C.

The optimal concentrations of the probes were determined empirically to be 0.15 μM for all, except for the IC probe, which was set at 0.06 μM because mammalian (human or primate) DNA was highly represented in the samples.

### 3.2. Analytical Sensitivity and Specificity

The LoDs obtained were estimated in the range of 0.6 to 3.01 parasites/µL for *Plasmodium* species, 0.0018 parasites/µL for *T. brucei*, and 2 microfilariae/mL in the case of *L. loa* (Table 2).

The analytical specificity was determined in the 102 positive blood samples. Only three blood DNA samples showed coinfections involving *P. falciparum* + *L. loa*. The specific PCRs (SnM-PCR and nFil-PCR) confirmed the results in all three cases, so it was determined that there was no cross-reactivity and that the specificity was 100%.

### 3.3. Intra-Assay Precision

Repeatability indicates the rate of variation of the method on the same sample when it is used repeatedly under the same conditions and in the same place, i.e., in this case, for three successive days using the same PCR machine (Table 3).

### 3.4. Inter-Assay Precision

Reproducibility refers to the percentage of variation in results when different people use the same method in different places or qPCR machines; in this case, it was performed using three different qPCR machines of the same brand (Rotor-Gene Q 6 plex (QIAGEN^®^) (Table 4).

### 3.5. Diagnostic Performance of the RT-PCR-bp Assay

A concordance of 99.1% (228/230) was achieved between the diagnostic results obtained with the RT-PCR-bp assay and the reference PCR methods of the Parasitology Reference and Research Laboratory (Table 5). Only two Filariae samples were not identified by the RT-PCR-bp assay.

The RT-PCR-bp assay showed sensitivity, specificity, PPV, and NPV values of 100% for *Plasmodium* spp., while for Filariae, these values ranged from 89.5% to 100% (Table 6). In the case of Trypanosomatidae, these values could not be statistically calculated due to the low sample number, although 100% of the samples gave the expected result.

Of the 58 stool DNA samples, 34 were positive according to RT-PCR-bp (58.6%), while, previously, only 27 (46.5%) samples tested positive in the study by Köster et al., 2021. In addition, the RT-PCR-bp assay allowed the identification of nine mixed infections that were missed by the reference PCR methods (Table 7). In Trypanosomatidae, three expected positive cases were negative, whereas four new positive cases were detected [17].

### 3.6. Operational Features

The estimated turnaround time to complete the process—from sample processing to the provision of results—was 3 h and 30 min for RT-PCR-bp. In comparison, the turnaround time for the reference methods was approximately the same for RT-PCR for Trypanosomatidae but significantly longer for conventional SnM-PCR and nested Filariae PCR (Table 8).

The hands-on time was notably shorter for RT-PCR-bp, involving only sample preparation, PCR setup, and result analysis, amounting to approximately 1 h. For the reference methods, the hands-on time increased significantly due to the need to prepare five separate PCRs, including the two conventional nested PCRs, and conducting the corresponding electrophoretic procedures, totalling approximately 5 h.

Regarding costs per sample, for RT-PCR-bp, the amplification kit, probes, and primers must be considered, assuming approximately EUR 6 per reaction. For RT-PCR on Trypanosomatidae, the cost is slightly lower, at EUR 5 per reaction, as it uses only two probes instead of four. For the two conventional nested PCRs, the cost is around EUR 2 per sample, including the amplification kits, dNTPs, primers, and electrophoresis components. Overall, the total cost for the reference methods amounts to EUR 9 per sample.

## 4. Discussion

Parasitic diseases, including malaria, human African trypanosomiasis, Chagas disease, and lymphatic filariasis—among other NTDs—continue to threaten over 1 billion people worldwide and are recognised as major global vector-borne diseases [3,27]. Despite the implementation of several control strategies and programmes in tropical countries, the complex life cycles of parasites, the existence of animal reservoirs, and the co-endemicity of multiple pathogens underscore the urgent need for new control approaches [28,29]. These approaches should incorporate MAS-based methods, instead of single detection systems such as rapid diagnostic tests or loop-mediated isothermal amplification (LAMP) [30], which offer highly sensitive and multiplexed screening for several pathogens in a single test [10,31].

This work presents a cost-effective and highly sensitive multiplexed tool (RT-PCR-bp) that can effectively and simultaneously detect *Plasmodium* spp., Trypanosomatidae, and filarial worms using specific fluorescence probes, allowing the detection of low parasite loads and coinfections. An asset of RT-PCR-bp is the incorporation of an internal reaction control for differentiating DNA extraction errors and amplification failures from true negative results. Furthermore, no unspecific amplifications or cross-reactions among the three groups of blood pathogens tested here were observed, proving the diagnostic usefulness of this method in detecting mixed infections.

RT-PCR-bp merges, in a single reaction, a previously designed RT-PCR method for Trypanosomatidae and a transformation of conventional PCR into an RT-PCR method for the detection of filarial worms with the design of a new specific probe [17]. Both assays are based on the amplification of the ssrRNA marker. In addition, two new primers and their corresponding specific probes have been designed based on the mitochondrial COI gene for the specific detection of *Plasmodium* spp.

According to the results obtained, while the analytical specificity was 100%, the only observed variation for the three groups of blood parasites was related to the reproducibility precision values due to each equipment’s inherent limitations in quantifying low parasite loads. It is important to note that these parameters are directly related to each piece of equipment and to their correct maintenance and calibration and not to the assay itself. However, both parameters demonstrated this method to be a reliable tool, with every sample correctly detected as positive for each group of parasites and each parasite load in every assay.

In addition, the statistical values showed that this method is comparable with our previously mentioned reference methods and the analytical sensitivity, with detection limits of 0.73–3.01 parasites/μL for *Plasmodium* spp. (depending on the species), 0.0018 parasites/μL for *T. brucei*, and 2 microfilariae/mL aligning with other established methods. For malaria, the LoD is comparable to those reported by other researchers using nested PCR, RT-PCR, and LAMP [14,15,16,30,32]. However, the RT-PCR-bp method demonstrates an LoD one logarithm lower for the detection of P. *ovale* (0.61 parasites/μL), compared to 0.21 and 2.3 parasites/μL in LAMP and RT-PCR, respectively [30,32]. These discrepancies may be attributed to differences in the *P. ovale* subspecies analysed, as previous studies did not differentiate between them [30]. Overall, this method showed a general sensitivity for *Plasmodium* spp., which corresponds favourably to other published RT-PCR methods with detection limits around 1 p/μL [33]. In Trypanosomatidae, various studies have reported LoDs ranging from 0.1 parasites/mL in the case of *T. brucei* [34,35] to 0.5 and 5 p/mL depending on the species [36,37] and genotype of *T. cruzi* [36,37,38], similar to those obtained with the current method (1.8 parasite/mL). For filariasis, the detection limit depends on the species, as shown by other authors [39]. For example, *Mansonella perstans* microfilaremia below 0.61 µf/mL is not detected, while parasitaemias over 2.5 µf/mL are detected, independently of the species [39]. The LoD of RT-PCR-bp was 2 µf/mL, which correlates with the previous study [39].

A key aspect of this method is the detection of mixed infections including two or more parasite groups. Initially, samples corresponding to mixed infections between groups were not included since these samples initially arrived at the Reference Laboratory to diagnose or confirm a single pathology, but when applying the method, three mixed infections of *Plasmodium* spp. and filarial worms were characterised. Likewise, 16% of coinfections were observed in the faecal samples of the chimpanzees. This shows the importance of multiple detection systems to perform the correct diagnosis of mixed infections, and, therefore, molecular methods are essential due to their greater sensitivity and specificity compared to microscopy [32]. The misdiagnosis of mixed infections may involve inadequate treatment, damaging the patient’s health and the control of the disease in endemic areas [32,40].

It is important to note that faecal samples are not optimal biological samples for detecting blood parasites such as *Plasmodium* spp., Filariae, or *Trypanosoma* spp., which are more accurately identified in blood samples [41]. However, other studies have shown the presence of these parasites in non-human primate faeces [42,43,44,45]. In this study, the value of using these samples lay in comparison with the previous published results by Köster et al. (2021) [17]. The new RT-PCR-bp method incorporates the Trypanosomatidae detection PCR method (RT-PCR-Tryp) used in that work, so no improvements were expected. On the contrary, in the case of Filariae, where a conventional nested PCR method has been changed to RT-PCR, the percentage of samples infected with Filariae has increased from 15.5% to 24.0%. In malaria, the increase is still greater, from 3.4% to 21.0%. This could be attributed not only to the switch from conventional PCR to RT-PCR but also to the change in target from ssrRNA to the mitochondrial COI gene [12,13]. All these new positives were verified using specific amplification methods, ruling out the possibility of false positives. Regarding operational characteristics, the time required to obtain results was significantly shorter with RT-PCR-bp compared to the reference methods. This efficiency is also evident in the reduced staff working time, with the reference methods requiring approximately 4 additional hours. Moreover, the cost of RT-PCR-bp is lower than the combined price of all the reference methods. Therefore, we recommend using the RT-PCR-bp assay as a screening tool, and, in the case of positive results, the corresponding PCR assays can then be employed to sequence the amplified fragments to identify the species involved in the infection.

The limitations of this study are related to the number of samples available for the different pathologies, especially in the cases of Trypanosomatidae and mixed infections. Another limitation is related to costs since the prices of amplification kits, probes, and primers vary between countries, and those reflected here may not be possible to extrapolate.

## 5. Conclusions

Overall, the RT-PCR-bp method presented here demonstrated high sensitivity, specificity, and cost-effectiveness as a multiple-analysis system capable of the simultaneous detection of *Plasmodium* spp., Trypanosomatidae, and filarial parasites. This approach enables the diagnosis of low parasite loads and coinfections, facilitating appropriate control measures for multiple etiological agents in a single test. Consequently, it contributes to timely and improved diagnosis and treatment.

## Figures and Tables

**Table 1 diagnostics-15-00620-t001:** Oligonucleotides used for the molecular detection of the parasites investigated in the present study. Specificity and final concentration values are indicated.

Primer	Sequence (5′–3′)	Specificity	Final Concentration (µM)
JM-U-0011F	CAAGTCTGGTGCCAGCA	Universal	0.2
JM-T-349R	CCAACAAAAGECGAAACGGTGGCC	Trypanosomatidae	0.2
JM-Fi-0015R	CAAGGTAAACTTGCTAGCCAC	Filariae	0.2
JM-P-COI2F	GGTGTGTACAAGGCAACAATAC	*Plasmodium* spp.	0.2
JM-P-COI1R	CATATAACGGTAAGAAGGTTCGC	*Plasmodium* spp.	0.2
IC-Forw	GAGCCGCCTGGATACCGC	Mammals	0.2
IC-Rev	GACGGTATCTGATCGTCTTC	Mammals	0.2
Tryp 681	TxRd–GCTGTTGCTGTTAAAGGGTTCGTAG–BHQ2	Trypanosomatidae	0.15
Fi 101	Cy5.5–GGTCCATYCATTGGATGAGAACT–BHQ2	Filariae	0.15
MALCOI 2	Cy5–ATTGGCACCTCCATGTCGTCTCAT–BHQ2	*Plasmodium* spp.	0.15
IC	Hex–TCGCTCTGGTCCGTCTTG–BHQ1	Mammals	0.06

BHQ1, Black Hole Quencher 1; BHQ2, Black Hole Quencher 2; Cy5, Cyanine-5 (Fluorescence lecture filter: Red); Cy5.5, Cyanine-5.5 (Fluorescence lecture filter: Crimson); Hex, Hexachlorofluorescein; IC, internal reaction control (Fluorescence lecture filter Yellow); TxRd, Texas Red Fluorescence lecture filter: Orange).

**Table 2 diagnostics-15-00620-t002:** Initial parasitaemia/microfilaremia levels and calculated limits of detection (LoDs) in the serially diluted samples.

Parasite	Initial Parasitaemia or Microfilaremia	LoD
*Plasmodium falciparum*	73,000 parasites/µL	0.73 parasites/µL
*Plasmodium vivax*	301 parasites/µL	3.01 parasites/µL
*Plasmodium ovale*	6110 parasites/µL	0.61 parasites/µL
*Plasmodium malariae*	107 parasites/µL	1.07 parasites/µL
*Trypanosoma brucei*	1800 parasites/µL	0.0018 parasites/µL
*Loa loa*	200 microfilariae/mL	2 microfilariae/mL

**Table 3 diagnostics-15-00620-t003:** Parasitic load obtained from the intra-assay precision analysis conducted in the present study. The values indicate the amount of parasites/µL for *Plasmodium* spp. (*P. falciparum* samples) and Trypanosomatidae (*T. brucei* samples) or the amount of microfilariae/mL for Filariae (*Loa loa* samples). Standard deviation and precision values are indicated.

Species	Parasitaemia	Day 1	Day 2	Day 3	Standard Deviation	Precision (%)
*Plasmodium* spp.	High	6.98 × 10^4^	7.08 × 10^4^	7.30 × 10^4^	1.64 × 10^3^	97.70
	Medium	7.02 × 10^3^	7.19 × 10^3^	7.30 × 10^3^	1.41 × 10^2^	98.03
	Low	4.84 × 10^1^	4.22 × 10^1^	5.36 × 10^1^	5.71 × 10^0^	88.13
Trypanosomatidae	High	1.84 × 10^4^	1.80 × 10^4^	1.84 × 10^4^	2.29 × 10^2^	98.75
	Medium	1.82 × 10^3^	1.74 × 10^3^	1.68 × 10^3^	7.26 × 10^1^	95.85
	Low	6.30 × 10^1^	8.10 × 10^1^	6.30 × 10^1^	1.04 × 10^1^	84.94
Filariae	High	1.19 × 10^5^	1.27 × 10^2^	1.19 × 10^2^	4.85 × 10^3^	96.00
	Medium	2.00 × 10^2^	2.03 × 10^2^	2.15 × 10^2^	7.94 × 10^0^	96.15
	Low	9.20 × 10^1^	6.60 × 10^1^	5.40 × 10^1^	1.94 × 10^1^	72.51

**Table 4 diagnostics-15-00620-t004:** Parasitic load obtained in the inter-assay precision analysis conducted in the present study. The values indicate the amount of parasites/µL for *Plasmodium* spp. (*P. falciparum* samples), Trypanosomatidae (*T. brucei* samples), and microfilariae/mL for Filariae (*Loa loa* samples). Standard deviation and precision values are indicated.

Species	Parasitaemia	M1	M2	M3	Standard Deviation	Precision (%)
*Plasmodium* spp.	High	2.59 × 10^3^	2.31 × 10^3^	2.01 × 10^3^	2.90 × 10^2^	87.41
	Medium	3.83 × 10^2^	4.30 × 10^2^	5.24 × 10^2^	7.18 × 10^1^	83.89
	Low	2.90 × 10^0^	5.01 × 10^0^	4.62 × 10^0^	1.12 × 10^0^	73.12
Trypanosomatidae	High	1.38 × 10^4^	1.37 × 10^4^	1.24 × 10^4^	7.81 × 10^2^	94.13
	Medium	2.52 × 10^3^	2.61 × 10^3^	1.38 × 10^3^	6.86 × 10^2^	68.40
	Low	1.41 × 10^2^	1.44 × 10^2^	2.54 × 10^2^	6.43 × 10^1^	64.16
Filariae	High	3.24 × 10^3^	2.83 × 10^3^	2.92 × 10^3^	2.15 × 10^2^	92.81
	Medium	3.12 × 10^2^	2.48 × 10^2^	3.53 × 10^2^	5.29 × 10^1^	82.61
	Low	1.72 × 10^1^	6.57 × 10^1^	2.24 × 10^1^	9.54 × 10^0^	61.99

M: Real-time PCR machine.

**Table 5 diagnostics-15-00620-t005:** Comparison of the diagnostic performance of RT-PCR-bp and the reference PCR methods for the detection of blood parasites in blood DNA samples (*n* = 230).

	Reference PCR Methods	RT-PCR-bp
*Plasmodium* spp.	119	119
*Plasmodium falciparum*	81	
*Plasmodium vivax*	13	
*Plasmodium malariae*	8	
*Plasmodium ovale*	8	
Mixed Infections	9	
Trypanosomatidae	9	9
*Leishmania infantum*	3	
*Trypanosoma brucei*	4	
*Trypanosoma cruzi*	2	
Filariae	33	31
*Mansonella perstans*	7	
*Loa loa*	24	
Mixed Infections	2	
Negative	69	71
Total	230	230

**Table 6 diagnostics-15-00620-t006:** Performance parameters of the RT-PCR-bp method developed in the present study; 95% confidence intervals (95% CIs) and Kappa index values are indicated.

	*Plasmodium* spp.		Filariae	
	Values (%)	95% CIs	Values (%)	95% CIs
Sensibility	100	94.0–100	93.9	80.4–98.3
Specificity	100	81.6–100	100	81.6–100
PPV	100	94.0–100	100	89.0–100
NPV	100	81.6–100	89.5	68.6–97.1
Kappa index	1.00	1.00–1.00	0.91	0.80–1.03

NPV, negative predictive value; PPV, positive predictive value.

**Table 7 diagnostics-15-00620-t007:** Comparison of the diagnostic performance of the RT-PCR-bp and the reference PCR methods for the detection of blood parasites in faecal DNA samples (*n* = 58).

	*Plasmodium* spp.	Trypanosomatidae	Filariae	Coinfected	Total Infected
RT-PCR-bp *n* (%)	12 (20.7)	17 (29.3)	14 (24.1)	9 (15.5)	34 (58.6)
Reference PCR method *n* (%)	2 (3.4)	16 (27.6)	9 (15.5)	3 (5.2)	27 (46.5)

Reference PCR method: study by Köster et al., 2021 [17].

**Table 8 diagnostics-15-00620-t008:** Estimated turnaround time to complete the process—from sample processing to the provision of results.

	RT-PCR-bp	SnM-PCR	nFil-PCR	RT-PCR-Tryp
Sample processing	1 h	1 h	1 h	1 h
First PCR setup	30 min	30 min	30 min	30 min
First PCR amplification	1 h 30 min	2 h	2 h	1 h 30 min
Second PCR setup	-	30 min	30 min	-
Second PCR amplification	-	1 h 15 min	1 h 15 min	-
Electrophoresis	-	30 min	30 min	-
Result analysis	30 min	15 min	15 min	30 min
Total	3 h 30 min	6 h	6 h	3 h 30 min

Sample processing includes DNA extraction and purification. PCR setup includes master mix and PCR preparation.

## Data Availability

All data are available upon request (jmrubio@isciii.es).

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
