# Peer review of "Development and Assessment of a Multiple-Analysis System for Diagnosing Malaria and Other Blood Parasite Infections in Humans and Non-Human Primates"

_diagnostics, 2025, doi:10.3390/diagnostics15050620_

Round 1
Reviewer 1 Report
Comments and Suggestions for Authors
Dear Authors,
I appreciate your efforts put in for development of the MAS test . I have concerns about the methodology and sample size , especially the positive samples which is extremely low.
Overall Comments:
This study aimed to describe and validate a new test system namely, the multipleanalysis system (MAS) for effective detection of numerous etiological agents of diseases i.e., Malaria, Chagas, Human African Trypanosomiasis, and Lymphatic Filariasis in a single test to facilitate their diagnosis in endemic countries where they are reported to co-exist.
The idea of development of a novel multiple analysis system (MAS) is appreciable and expected to be of great value as a diagnostic test for individual patients as well as a screening test for endemic populations. Considering the immense importance of a diagnostic and screening test for patients infected with these diseases, it is of utmost importance that the methodology used for its evaluation is appropriate and scientifically sound. For evaluation of the basic parameters like sensitivity, specificity and PPV and NPV , it is important that a minimum sample size is used to arrive at meaningful conclusions. In order to streamline this process the STARD guidelines are usually considered as a benchmark for reliable results.
The authors have taken an extremely low and statistically invalid sample size which appears to be biased , especially the positive samples (n=3)for testing the sensitivity, based on which it is not possible to arrive at the diagnostic value and detection limit of the test developed by the authors. The values for high, medium and low positivity also appear to be arbitrary without any specific values for corresponding parasite count against a gold standard. Therefore, the conclusion drawn by the authors, ‘the MAS test developed is cost-effective and highly sensitive multiplexed tool (RT-PCR-bp) capable of performing a simultaneous detection for blood parasitic diseases using specific fluorescence probes, enabling the diagnosis of low parasite loads and co-infections’, does not have enough scientific proof and supporting data in the paper to authenticate or validate the claim. The predictive values cannot be calculated on such a low sample size and usually are prevalence based. Other parameters like ‘Inter and Intra-assay precision’ also fall short of an ideal and accepted methodology used for development of new diagnostic tests.
Further, it is not clear whether the authors expect this new MAS diagnostic test to be accepted based on this study or they have further plans to work on it. The utilization of diagnostic tests in patient care settings must be guided by evidence. Sensitivity and specificity are essential indicators of test accuracy and allow to determine the appropriateness of the diagnostic tool with the proper level of confidence in the results derived from known sensitivity, specificity, positive predictive values (PPV), negative predictive values (NPV) and other performance parameters are also equally important for development of a new diagnostic test.
In view of the stated shortcomings, it might be useful for the authors to work more on the evaluation of this diagnostic tool following the accepted methodology and norms as well as statistically valid minimum sample size requirements. The suggestion of a relevant statistician would be useful for designing the study appropriately. Though I am not sure but would still request the authors to rethink if it would help them in any way to publish this study, after appropriate revisions, as an initial laboratory test report on the diagnostic potential of the MAS test developed by them with plans for its further evaluation and development. This might help them in documenting their efforts in the first instance and allow them sufficient time to work on the MAS test further.
Specific detailed comments on various sections of the paper are not being offered separately in view of the comments offered above.
Best wishes .
Comments on the Quality of English Language
The english language should be improved while revising/rewriting the paper for clarity and the background, work carried out on the subject previously, methodology , results etc presented clearly for benefit of the researchers and readers.
Author Response
1. Summary |
|
|
|||||||||
We are very grateful for the comments made by reviewer 1 and will try to resolve and clarify them in the best possible way. However, we consider that some of your comments do not fit the reality of our manuscript. Please find the detailed responses below of the extracted comments from the text, and the corresponding revisions/corrections highlighted and mark in red in the re-submitted files.
Nothing else to clarify.
|

Reviewer 2 Report
Comments and Suggestions for Authors
Dear Editor
The objective of the present study was to describe and validate a new highly sensitive and specific multiple-analysis system that can effectively detect numerous etiological agents in a single test", and there are many structural, language errors that should be corrected.
Major comments:
- The title of the manuscript is too long and should be reworded. I request to modify it to "Development and assessment of a multiple analysis systems for diagnosing malaria and other blood parasite infections in humans and nonhuman primates"
- The abstract section needs substantial rewording, e.g.
Line 26: "Tackle this disease lies in a correct diagnosis, sensitive, specific, and fast" replace with "Tackling this disease requires an accurate diagnosis that is sensitive, specific, and rapid".
Line 29: "A total of 230 human blood samples were evaluated retrospectively for parasite characterization and a total of 58 stool samples from Non-Human Primates (NHP) "rephrased to "A total of 230 human blood samples were assessed retrospectively for parasite characterisation, as well as 58 stool samples from Non-Human Primates (NHP)"
- A distinct section should be added to include the statistical analysis.
- Rewording the discussion and conclusion is necessary.
Major comments:
Which travelers or immigrants' nationality do you specify for sample collection?
Minor comments:
Line 33: "1,8 parasite/ml" replace with " 1.8 parasite/ml"
Line 36: "93,9%," replace with " 93.9%, "
Line: 45 " Many tropical diseases are caused by parasites and transmitted by vectors. Among 45 them is malaria, with 608,000 deaths in 2022" replace with " Many tropical diseases are transmitted by vectors and are brought on by parasites. Malaria is one of 45, and it killed 608,000 people in 2022.
Line 76: "The validation of the developed method was carried out by comparing the obtained results against accredited laboratory methods for the detection of these parasites including" rephrased to "The developed method was validated by comparing the obtained results against accredited laboratory methods for detecting these parasites, including……."
Line 124: "for the characterization of" replace with " to characterise"
Line 126: " For the identification of" replace with " to indentify"
Line 160: " a RT-PCR' replace with " an RT-PCR"
Comments on the Quality of English Language
The article's language does not adhere to the accepted grammar rules for producing scientific reports.
Author Response
1. Summary |
|
|
|||||||||
Thank you very much for taking the time to review this manuscript. Please find the detailed responses below in black Italics and the corresponding revisions/corrections highlighted/and in red, in the re-submitted files.
None
|
Reviewer 3 Report
Comments and Suggestions for Authors
In this paper, the authors describe the RT-PCR method for the multiple detection of Plasmodium malaria, Trypanosomatidae, and Filariae parasites. While the method is potentially interesting, it suffers from some incomplete methodological descriptions in the actual text. Revision is required.
Critics:
- In the actual title, it looks like both blood and stool were tested in both humans and non-human primates, which was not done. Fix this misinformation in the title (perhaps by indicating "respectively").
- In the abstract, line 35, specify that the kappa coefficient reaches 100% only for Plasmodium.
- In the abstract, line 36, define NPV when first used.
- In the introduction, better explain why stool samples are important for testing for malaria and why it’s included in this study. It’s known that in malaria patients, Plasmodium DNA traces in stool are rare and seem to have no diagnostic value.
- General comment: unfortunately, only a few samples with mixed infections were analyzed and verified by the presented method.
- In the methods section, Table 1: The primers for mammals' genes are reported without any comment in the text (such as the use of these primers, which mammalian gene, the gene specificity, etc.).
- In the methods section, COI primers, which are the true novelty of this study, are mentioned for the first time. Please underline this new gene for PCR in the introduction and possibly in the abstract.
- In the methods section, the part about PCR probes is completely absent. Under Table 1, in lines 135-136, some abbreviations are accumulated without any explanation, e.g. matching with primers, concentrations, and other important information useful for reproducing the test.
- In the methods section, it is not clearly described if all primers should be mixed in the same tube with the sample, or if more tubes with the same sample are needed for this "multiple detection system." Please describe this part carefully, as it is the only new aspect of RT-PCR-bp compared to the standard RT-PCR tests with conventional Plasmodium, Trypanosoma, or Filariae primers.
- In the methods section, lines 140-141: The text mentions "...the corresponding amount of primers and probes (Table 1)," but no probes and their amounts are reported in Table 1. Generally, in chapter 2.4, the probe-wavelength pairing is not specified.
- In the methods section, line 157: Before the citation [12], another citation appears to be missing.
Results:
- Specify better, based on which sample number and parasitemia range the LoD data were obtained. In the current Table 2, only one value of initial parasitemia is indicated for each parasite species. Even in line 167, the processed samples of “each” patient are mentioned, thus a certain range of parasitemia was available. Another perplexity is the high parasitemia initial levels reported in Table 2. Small parasitemias are much more interesting in the field.
- Tables 3 and 4: DNA concentrations are reported without the units of measurement.
- Results, line 229: “Different qPCR machines” are mentioned. Which machines exactly?
- Results, lines 249-253: Stool analysis. The authors observed more positive samples with their RT-PCR-bp method compared to conventional methods and interpreted this as higher sensitivity. However, this could be a “false positive” result. Please consider and discuss this.
- Results, line 264: “The hands-on time was notably shorter for the RT-PCR-bp, involving only sample preparation, PCR setup, and result analysis, amounting to approximately 1 hour.” However, in Table 8, the total time, including result analysis, is 3 hours and 30 minutes. Please, fix this incongruence.
- References 1 and 2: Add the web link.
Author Response
Thank you very much for taking the time to review this manuscript. Please find the detailed responses below in black Italics and the corresponding revisions/corrections highlighted/and in red, in the resubmitted files.
3. Point-by-point response to Comments and Suggestions for Authors
|
Comments 1: In the actual title, it looks like both blood and stool were tested in both humans and non-human primates, which was not done. Fix this misinformation in the title (perhaps by indicating "respectively").
|
Response 1: Thank you for pointing this out. We have changed the title following the recommendation of reviewer 2. “Development and assessment of a multiple analysis systems for diagnosing malaria and other blood parasite infections in humans and nonhuman primates”.
|
Comments 2: In the abstract, line 35, specify that the kappa coefficient reaches 100% only for Plasmodium. Response 2: Thank you for the comment, but we don’t fully agree as it’s written in the abstract manuscript: “The sensitivity, specificity, predictive values, and kappa coefficient reached almost 100%, except for the sensitivity for Filariae, which dropped to 93,9%, and its NPV to 89.5%” (Page/Line:1/34) and it is also showed in table 6 (Page/Line:7/255). This Performance parameters are only calculated for Plasmodium spp. and Filariae, thus, we consider that in the abstract, it is clear enough and it corresponds with the exact numbers described in table 6.
Comments 3: In the abstract, line 36, define NPV when first used. Response 3: Thank you for the comment. As requested, we have changed it for “Negative Predicted Value”.
Comments 4: In the introduction, better explain why stool samples are important for testing for malaria and why it’s included in this study. It’s known that in malaria patients, Plasmodium DNA traces in stool are rare and seem to have no diagnostic value.
Comments 5: General comment: unfortunately, only a few samples with mixed infections were analyzed and verified by the presented method. |
Response 5: Yes, we agree. It is difficult to obtain such type of samples. In the initial validation, the use of mixed infection chimeras was proposed, but this was discarded because it was not easy to complete a range of all possibilities.
Comments 6: In the methods section, Table 1: The primers for mammals' genes are reported without any comment in the text (such as the use of these primers, which mammalian gene, the gene specificity, etc.). Response 6: Thank you for your comment. We included in the introduction a brief sentence about the genes target: “Primers and probes were designed in the small subunit ribosomal RNA (ssrRNA) gene, except for Plasmodium spp., for which the novel target was the Cytochrome Oxidase Subunit 1 (COI), due to its higher sensitivity [12, 13]” Page/Line: 2/77 Furthermore, in the methodology section we have included the next sentence: “For the internal reaction control, previously designed primers, targeting the human ssrRNA were used [14, 15].” And new references had been included.
Comments 7: In the methods section, COI primers, which are the true novelty of this study, are mentioned for the first time. Please underline this new gene for PCR in the introduction and possibly in the abstract.
Comments 8: In the methods section, the part about PCR probes is completely absent. Under Table 1, in lines 135-136, some abbreviations are accumulated without any explanation, e.g. matching with primers, concentrations, and other important information useful for reproducing the test. Response 8b: Regarding concentration, in order to clarify, we will substitute in table 1, Concentration (µM) for Final Concentration (µM). Page/Line: Table 1, 3/146 Response 8c: Regarding the comment of abbreviation matching with primers, we would like to clarify that those abbreviations are only the names of the primers and probes designed by us, such as: “JM-U-0011F, JM-Fi-0015R, etc..”
Comments 9: In the methods section, it is not clearly described if all primers should be mixed in the same tube with the sample, or if more tubes with the same sample are needed for this "multiple detection system." Please describe this part carefully, as it is the only new aspect of RT-PCR-bp compared to the standard RT-PCR tests with conventional Plasmodium, Trypanosoma, or Filariae primers. Response 9: We consider this point of interest and thought it was clear enough. In order to clarify we incorporate in the introduction a sentence: “The method is carried out in a single tube using different primers and specific probes…” Page/Line: 2/75 and in Methods section we have substituted in the methods section “The RT-PCR-bp reaction mix consisted of 1x Quantimix HotSplit (Biotools, Madrid, Spain) which contained the buffer, the polymerase and dNTPs, the corresponding amount of primers and probes (Table 1), and 5 µl of template DNA in a final reaction volume of 20 µl.” For “The RT-PCR-bp reaction mix consisted of 1x Quantimix HotSplit (Biotools, Madrid, Spain) which contained the buffer, the polymerase and dNTPs, the corresponding amount of primers and probes all mixed together in a single tube (Table 1), and 5 µl of template DNA in a final reaction volume of 20 µl.”
Comments 10 In the methods section, lines 140-141: The text mentions "...the corresponding amount of primers and probes (Table 1)," but no probes and their amounts are reported in Table 1. Generally, in chapter 2.4, the probe-wavelength pairing is not specified. Response 10: Thank you for your comment. We considered that this point was clear since the final concentration of probes and primers are on the table 1 and final volume of the PCR reaction is included in the methodology section and is 20µl.
Comments 11: In the methods section, line 157: Before the citation [12], another citation appears to be missing. Response 12: Thank you for your point. We agree completely and we have delete the coma and change it as follows in the manuscript: ” Validation of the method was performed by direct comparison of obtained results with those from accredited PCR protocols in the laboratory. The reference method for Plasmodium detection was a nested multiplex Malaria PCR (SnM-PCR) that identified the four Plasmodium species that normally infect humans (P. falciparum, P. malariae, P. ovale, and P. vivax) [14, 15].” Considering the addition of a new reference [14] as mention in the file Letter to the corresponding editor via email. Page/Line: 4/164
RESULTS
Comments 1: Specify better, based on which sample number and parasitemia range the LoD data were obtained. In the current Table 2, only one value of initial parasitemia is indicated for each parasite species. Even in line 167, the processed samples of “each” patient are mentioned, thus a certain range of parasitemia was available. Another perplexity is the high parasitemia initial levels reported in Table 2. Small parasitemias are much more interesting in the field. Response 1: Thank you for your point. They are not really high parasitemias, in the case of P. falciparum, which has the highest initial level, it is 1.46%, and for P. vivax, for example, it is 0.0006%. This proves that this methodology is really very capable of identifying very low parasitemias, which, as the reviewer says, is what can occur in the field in asymptomatic cases.
Comments 2: Tables 3 and 4: DNA concentrations are reported without the units of measurement. Response: At this point we have made an error in the title of the table which has led to confusion for the reviewer. We really want to indicate parasite loads here and not DNA concentration, so we have changed the titles of the tables, replacing DNA concentration with Parasite loads expressed as parasites/µl for Plasmodium spp. (P. falciparum samples) and Trypanosomatidae (T. brucei samples) or the amount of microfilariae/ml for Filariae. “Table 3. Parasitic Load …. Page/Line: 6/233 “Table 4. Parasitic Load …. Page/Line: 6/242
Comments 3: Results, line 229: “Different qPCR machines” are mentioned. Which machines exactly? Response 3: The three different machines used belongs to the same brand, but simply are different equipment. In order to clarify, we will add in the manuscript the following: “…three different qPCR machines from the same brand (Rotor-Gene Q 6 plex (QIAGEN®) (Table 4). Page/Line: 6/243
Comments 4: Results, lines 249-253: Stool analysis. The authors observed more positive samples with their RT-PCR-bp method compared to conventional methods and interpreted this as higher sensitivity. However, this could be a “false positive” result. Please consider and discuss this. Response 4: This is a good point. We did test each positive result with it corresponding specific PCR. We have added this in the discussion by adding the following sentence: “All these new positives were verified using specific amplification methods, ruling out the possibility of false positives..”
Comments 5: Results, line 264: “The hands-on time was notably shorter for the RT-PCR-bp, involving only sample preparation, PCR setup, and result analysis, amounting to approximately 1 hour.” However, in Table 8, the total time, including result analysis, is 3 hours and 30 minutes. Please, fix this incongruence. Response 5: We don’t fully agree with this comment. It is not the same hands-on or turn-around because the first indicates the technician's work time (hands-on) (30 for PCR setup and 30minutes for analysis results ) and the other is the time of performing the technique including the PCR waiting time (1h30PC PCR amplification +1h for sample processing, +30 minutes for reading and analysis of results, totaling 3h and 30minutes).
Comments 6: References 1 and 2: Add the web link. Response 6: We appreciate this comment and we will add the corresponding links.
Añadir en la discusión que no son falsos positivos porque se han comprobado con otras PCR especificas tal como se ha hecho con las muestras humanas de sangre.
|
4. Response to Comments on the Quality of English Language |
Point 1: The English is fine and does not require any improvement. |
|
l5. Additional clarifications |
None
Round 2
Reviewer 1 Report
Comments and Suggestions for Authors
Dear authors,
I accept the clarifications provided by you and feel that the paper has been thoroughly revised for publication , except for some grammatical mistakes that can be removed by another revision or editing.
Best wishes.
Reviewer 2 Report
Comments and Suggestions for Authors
Greetings, Author
I appreciate your response; every comment was addressed appropriately.